# Meningioma–Brain Crosstalk: A Scoping Review

**DOI:** 10.3390/cancers13174267

**Published:** 2021-08-24

**Authors:** Josefine de Stricker Borch, Jeppe Haslund-Vinding, Frederik Vilhardt, Andrea Daniela Maier, Tiit Mathiesen

**Affiliations:** 1Department of Neurosurgery, Rigshospitalet, Copenhagen University Hospital, 2100 Copenhagen, Denmark; Josefine.de.stricker.borch@regionh.dk (J.d.S.B.); andrea.maier@regionh.dk (A.D.M.); tiit.illimar.mathiesen@regionh.dk (T.M.); 2Department of Cellular and Molecular Medicine, Panum Institute, University of Copenhagen, 2200 Copenhagen, Denmark; vilhardt@sund.ku.dk; 3Department of Pathology, Rigshospitalet, Copenhagen University Hospital, 2100 Copenhagen, Denmark; 4Department of Clinical Medicine, University of Copenhagen, 2200 Copenhagen, Denmark; 5Department of Clinical Neuroscience, Section for Neurosurgery, Karolinska Institutet, 171 77 Stockholm, Sweden

**Keywords:** meningioma, inflammation, immune cells, tumour microenvironment, microglia, macrophage

## Abstract

**Simple Summary:**

Meningioma is the most common primary intracranial tumour. However, the histopathological diagnosis remains simplistic and to some extent insufficient compared to other brain tumours. Surgery is the primary treatment, and radiation therapy is secondary treatment for tumours that recur. Traditional chemotherapy has so far been ineffective, as these tumours are resistant, and research on meningioma biology is lacking compared to other tumour types. The tumoral microenvironment (TME) plays a key role in understanding various cancers. In meningiomas, however, the TME is poorly understood. It is unknown how the brain immune cells contribute to meningioma behaviour and aggressiveness, and the relationship between meningioma cells and TME involved in treatment resistance also needs to be investigated. It is therefore necessary to explore if the literature holds any evidence regarding meningioma–brain crosstalk in order to identify the gaps of knowledge. The sparse amount of available literature on the subject necessitates a scoping review approach.

**Abstract:**

**Background**: In recent years, it has become evident that the tumoral microenvironment (TME) plays a key role in the pathogenesis of various cancers. In meningiomas, however, the TME is poorly understood, and it is unknown if glia cells contribute to meningioma growth and behaviour. **Objective**: This scoping review investigates if the literature describes and substantiates tumour–brain crosstalk in meningiomas and summarises the current evidence regarding the role of the brain parenchyma in the pathogenesis of meningiomas. **Methods**: We identified studies through the electronic database PubMed. Articles describing glia cells and cytokines/chemokines in meningiomas were selected and reviewed. **Results**: Monocytes were detected as the most abundant infiltrating immune cells in meningiomas. Only brain-invasive meningiomas elicited a monocytic response at the tumour–brain interface. The expression of cytokines/chemokines in meningiomas has been studied to some extent, and some of them form autocrine loops in the tumour cells. Paracrine interactions between tumour cells and glia cells have not been explored. **Conclusion**: It is unknown to what extent meningiomas elicit an immune response in the brain parenchyma. We speculate that tumour–brain crosstalk might only be relevant in cases of invasive meningiomas that disrupt the pial–glial basement membrane.

## 1. Introduction

Meningiomas are the most common primary intracranial tumour [1]. According to WHO 2016 guidelines, meningiomas are classified as either grade I, grade II (atypical) or grade III (malignant), depending on mitotic rate, brain invasion and specific histological features. Treatment strategies span from watchful waiting to complete/partial surgical resection and/or radiation therapy [1]. Most meningiomas are benign and associated with a good prognosis [2]. However, the treatment options for recurrent or unresectable tumours are very limited, as they are resistant to traditional chemotherapy [3]. Recent research on the treatment of various cancer types focuses on immune checkpoint inhibitors and other types of immunotherapy [4]. The tumoral microenvironment (TME), the stromal contribution and the role of immune cells are all critical regulators of cancer progression in the brain. The TME is created by tumour cells, which orchestrate the cellular and molecular events that take place in the surrounding tissue [5]. The TME contains a substantial proportion of non-neoplastic cells, such as pericytes; endothelial cells; fibroblasts; immune cells; and cells distinctive for brain tissue, including microglia, astrocytes and stem-like cells [6,7,8]. These cells create a complex interplay of tumour–immune cell interactions in the TME [4]. In search of efficient immunotherapy for glioblastomas, recent research has unfolded the interesting mechanism of crosstalk between glioblastoma cells and glial cells from the brain parenchyma, which impacts tumour malignancy and serves as a basis for future potential therapeutic strategies. This crosstalk can be mediated by either soluble messengers across the glia limitans or by direct cell interactions. The interaction between the blood–brain barrier (BBB), the glia limitans and meningiomas has not yet been studied, and the TME in meningiomas is poorly understood. Thus, it remains unknown to what extent a possible reciprocal tumour–brain crosstalk takes place in meningiomas and how it affects the pathophysiology [9]. We (unpublished results) and others [10,11,12] have studied inflammatory cells in meningiomas, but the brain parenchymal responses have not been reviewed. In this review, we aim to analyse the literature on meningioma cells and glial cells (microglia and astrocytes) to describe tumour–brain crosstalk in meningiomas. On the basis of knowledge on tumour–brain crosstalk in other primary brain tumours [3], we address two aspects of crosstalk in meningiomas:(1)Attraction and activation of glial cells from the parenchyma to the tumour site;(2)Reciprocal exchange of substances and cell–cell interactions between glial cells and tumour cells that impact on meningioma growth and behaviour.

We analyse these qualities in order to explore if the literature holds any evidence regarding meningioma–brain crosstalk with emphasis on interactions mediated by soluble messengers. In addition, we identify gaps of knowledge, necessary to elaborate the role of parenchymal glial cells in meningioma pathogenesis. The complex and diverse nature of tumour–brain crosstalk coupled with the sparsity of the literature necessitate a scoping review approach.

## 2. Methods

A broad systematic search strategy was used to maximise the capture of relevant information on the topic. Search terms included the cells of interest (meningiomas, astrocytes and microglia) and molecules that mediate reciprocal tumour–brain crosstalk in other primary brain tumours (cytokines and chemokines) [13]. On the 16th of February 2021, a systematic search was carried out through an online search in PubMed using the MeSH database. The MeSH terms “meningioma”, “astrocytes”, “microglia”, “chemokines”, “receptors, chemokine” and “cytokines” were used. The MeSH term “receptors, cytokine” did not add any articles to the search and was therefore not included. Thus, our search was as follows:


*(meningioma[MeSH Terms]) AND (astrocytes[MeSH Terms]) OR (meningioma[MeSH Terms]) AND (microglia[MeSH Terms]) OR (meningioma[MeSH Terms]) AND (receptors, chemokine[MeSH Terms]) OR (meningioma[MeSH Terms]) AND (“chemokines”[MeSH Terms]) OR (meningioma[MeSH Terms]) AND (cytokines[MeSH Terms])*


Additional relevant references from the obtained literature were reviewed. With exception of case reports, clinical trials and letters to the editor, we included all English-written studies based on original data for title and abstract screening, regardless of year of publication. As no articles directly assessed the topic of our review, our inclusion and exclusion criteria were dynamic, as new ideas emerged. Articles related to peritumoral oedema, tumour angiogenesis, systemic inflammatory markers and tumour genetics were excluded. A PRISMA flowchart was made including reasons for exclusion of the literature (Appendix A).

## 3. Results

### 3.1. Microglia/Macrophage Infiltration in Meningiomas

Several studies reported the infiltration of tumour-associated microglia/macrophages (TAMs) in meningiomas [9,11,12,14,15]. The degree of infiltration was variable, but a majority of the studies reported them as the most abundant immune cell type in meningiomas with a mean content spanning from 18% to 44% [9,15,16,17]. Immunohistochemical studies identified TAMs as single cells or scattered groups of cells, with higher density near blood vessels or hypoxic and necrotic areas [9,14]. The morphology showed significant heterogeneity, including round, rod-shaped, ameboid and ramified TAMs [15]. Studies reported conflicting results regarding the association between the number of TAMs, tumour size and WHO grade [9,11]. The mechanisms of TAM infiltration were poorly investigated with only a few studies. The chemokine monocyte chemoattractant protein 1 (MCP-1) is expressed in meningiomas, and the degree of TAM infiltration correlated with MCP-1 expression. Thus, MCP-1 might play a role in the mechanism of TAM infiltration [18,19]. As microglia and blood-derived macrophages share similar immune regulatory mechanisms and express several common surface markers, it remains a challenge to discriminate between the two cell types [20]. There are emerging markers to distinguish microglia (P2Ry112, TMEM119) from macrophages (CD163, CD68), but further studies are needed to evaluate their reliability in humans [21,22,23]. Therefore, it is not clear whether the TAMs in meningiomas are blood-derived macrophages, migrated microglia or a combination of the two.

### 3.2. Classical (M1) or Alternative (M2) Microglia/Macrophage Activation in Meningiomas

TAMs differentiate into a classical (pro-inflammatory, M1) or alternative (anti-inflammatory, M2) phenotype. However, polarisation more likely represents an overlapping continuum of phenotypes rather than separate categories [24,25,26]. The polarisation of TAMs towards an M1 or M2 phenotype depends on the cytokine milieu and the local TME. Conversely, the polarisation state reciprocally affects the TME, as M1/M2 TAMs secrete pro- and anti-inflammatory substances, respectively. The polarisation state of TAMs can be recognised by detecting surface proteins and secreted substances associated with the M1/M2 phenotypes (Figure 1) [27]. Very few studies have investigated the immune phenotype of TAMs in meningiomas, but emerging research points to an important association between the polarisation state of TAMs and tumour progression [28]. Proctor et al. [9] found that infiltration of TAMs increased with WHO grade, and the M1:M2 cell ratio was significantly decreased in WHO grade II compared to grade I tumours, but the relationship between the post-operative outcome and the TME dynamics in regard to the extent of inflammation remains to be explored.

In meningiomas carrying the monosomy 22/del(22q), TAMs are polarised towards the pro-inflammatory M1 phenotype [17], and the M1/M2 ratio generally seems to decline with increasing WHO grade and in recurrent meningiomas also [9,29]. Thus, while evidence regarding the prognostic value of the number of TAMs is conflicting, the M1/M2 ratio might have significant importance in meningioma growth and behaviour.

**Figure 1 cancers-13-04267-f001:**
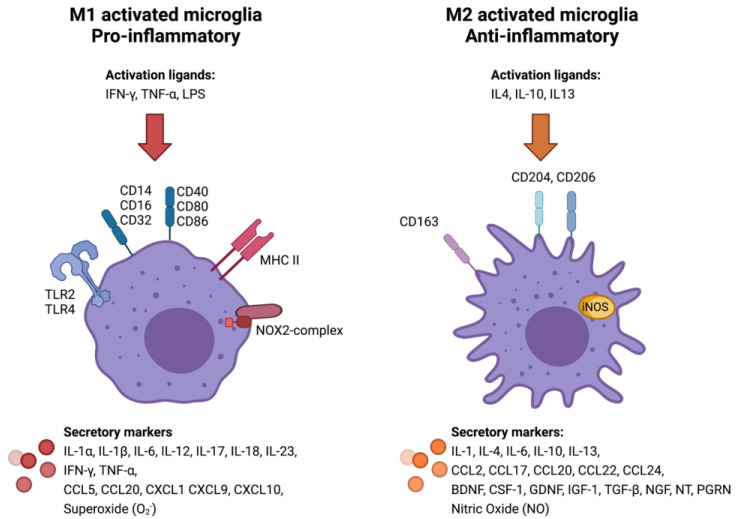
Surface and secretory markers associated with M1/M2 polarisation state of microglia [13,27,30,31,32,33].

### 3.3. Glial Response at the Tumour–Brain Interface

According to the small available number of studies, there were variable degrees of gliosis in the brain parenchyma adjacent to meningiomas [34,35,36,37]. The brain–meningioma interface of benign meningiomas is demarcated by a pial–glial basement membrane, which is typically absent in atypical and anaplastic variants [36]. Grund et al. 2009 [35] found a microglial/macrophagic response at the tumour–brain interface in a subset of invasive meningiomas, which correlated with the degree of malignancy and with loss of the pial–glial basement membrane. The response was of two different kinds; the majority of cases had a wall of mononuclear cells separating the tumour from the brain, while a minor subset showed activated microglia in the brain parenchyma [35]. Astrocytes could be found in the subjacent compressed brain tissue or deep in the tumour between the lobules, probably due to the entrapment of vessels with the surrounding astrocytes [36,37]. The astrocytes eventually appeared to disappear from the tumour–brain interface [34], as invasive meningiomas progressively disrupted the pial–glial basement membrane.

### 3.4. Cytokine Expression in Meningiomas

A substantial number of studies have investigated the expression of cytokines in meningiomas with different methods, including rt-PCR, immunohistochemistry, Western blot and flow cytometry. From these studies, it is evident that meningiomas express a wide range of cytokines, chemokines and growth factors with pro- and anti-inflammatory properties. The roles of these substances in the course of meningioma progression remain largely unexplored. Table 1 summarises the current knowledge on cytokine/cytokine receptor expression in meningiomas.

### 3.5. Microenvironmental Impact on Meningioma Growth and Behaviour

Cytokines can stimulate or inhibit meningioma proliferation in vitro [43,45,54,59,60,61]. While TGF-ß [62], IL-6 [45,61] and OSM [43] inhibit meningioma cell growth, SDF-1 [54,60] and CXCL16 [59] increase proliferation in cultured meningioma cells. In addition, the expression of cytokines appeared to correlate with malignancy grade [46,63] and tumour recurrence [53,64,65,66,67,68]. TGF-ß levels seemed to decline with increasing WHO grade [63]. In contrast, G-CSF, GM-CSF and osteopontin (OPN) levels correlated positively and significantly with increasing WHO grade [46,64]. In terms of tumour recurrence, lower TGF-ß expression and higher expression of OPN, HGF/SF, MIP and MMP-9 were found to correlate significantly with the recurrence of meningiomas [53,64,65,66,67]. However, the independent predictive value of HGF/SF expression was not associated with tumour recurrence according to Kärja et al. [68]. In summary, the cytokine milieu modulates tumour cell growth and varies among different malignancy grades in recurrent vs. non-recurrent meningiomas.

### 3.6. Sources and Targets of Cytokines in Meningiomas

The majority of the studies on cytokines in the meningioma TME focused on the effect on tumour cells upon receptor binding, and, thus, little is known regarding other potential cellular sources and targets of cytokines in the TME. Some cytokines/chemokines, e.g., IL-6, CXCL12 and TGF-ß, show a formation of autocrine loops in meningioma cells. Such loops have been demonstrated as a signalling mechanism that impact tumour behaviour (Figure 2) [43,45,59,61]. Some chemokine/chemokine receptor pairs have also been detected in endothelial cells and in TAMs [58], implying that paracrine signalling also takes place in the TME. It has not been investigated whether microglia and astrocytes contribute to the cytokine milieu in meningiomas or if and how the TME reciprocally affects these glia cells.

## 4. Discussion

### 4.1. Meningioma–Brain Crosstalk: Current Research Knowledge Base

This study provides an overview of the existing evidence of reciprocal crosstalk between tumour cells and the brain parenchyma in meningiomas. Meningiomas express several factors, which are known to have chemotactic properties on microglia in glioblastoma multiforme (e.g., CCL2, CX3CL1, SDF-1, G-CSF and GM-CSF) [18,26,56,69]. We inferred that brain invasion in meningioma is mandatory to elicit an immune response in the brain parenchyma; however, direct invasion is only seen in a minor subset of cases. Monocytes infiltrate all meningiomas diffusely [9], but only a subset of the brain-invasive meningiomas had a monocytic response at the tumour–brain interface. Meningiomas are located outside the blood–brain barrier [70] and usually supplied from the external carotid artery [71], and, thus, it seems plausible that the diffuse monocytic infiltrates in meningiomas represent blood-derived macrophages, while the monocytic response at the tumour–brain interface may comprise microglia. Tumour-associated astrocytes, however, disappeared during the course of brain invasion [34]. This is in contrast to a wide range of other CNS pathologies, such as stroke and glioblastomas, which are associated with reactive astrogliosis with increased astrocyte proliferation and GFAP levels [13,72,73]. In specimens with brain invasion, focal absence of a pial basal membrane correlated with the absence of subpial astrocytes [34]. The authors interpreted the finding as “astrocytic disappearance” after invasion and suggested that astrocyte survival was dependent on adherence to an intact basal membrane. Loss of astrocytes would then be a result of a degraded basal membrane. In analogy, breaches of the basal membrane are considered hallmarks of malignancy in cancers [74]. Degradation of the basal membrane could be mediated by matric metalloproteases, which are expressed by tumour cells in invasive and aggressive meningiomas [75,76]. Both the known co-expression of MMP-9 with MIF-1 [65] and immune regulatory T-cells would further allow brain invasion by attenuation of the microglial response [77]. Although brain invasion in meningioma does not necessarily lead to the diagnosis of WHO grade III meningioma, brain infiltration still appears to be associated with aggressive tumour behaviour [2,10,78,79,80].

The pia limitans (pial–glial basement membrane) is found in the perivascular space, composed of astrocytic end feet connected with inducible tight junctions and constitute the last frontier before the brain parenchyma [81,82]. Joost et al. 2019 [83] have investigated the pia limitans and found that the microglia participate in the formation of the glia limitans in human post-mortem brain tissue. Since microglia are reported to contribute to the formation of the perivascular glia limitans under inflammatory conditions, it renders them ideally positioned to control or respond to leukocyte infiltration and soluble signalling [83]. The pia limitans could very well be compromised by meningioma, especially invasive WHO grade II and III, but is yet unexplored.

In summary, the few available studies indicate that the pial–glial basement membrane hinders a microglial response in the brain parenchyma, but it may simultaneously support and ensure survival of the subpial astrocytes.

The TAMs were predominantly of the pro-tumoral M2 type, and the M1/M2 ratio decreased with increasing WHO grade [9,84]. This suggests a reciprocal crosstalk between tumour cells and TAMs [84]. The phagocytes at the tumour–brain interface, which are likely to represent resident microglia, have not been studied or well characterised yet. Thus, the activation state of these cells remains unknown.

Microglia/astrocyte–tumour interactions in primary brain tumours are mediated by several factors, including cytokines and chemokines and neurotrophic, morphogenic and metabolic factors [13]. These factors are expressed in meningiomas, but the roles of these substances were mainly extrapolated from research conducted on glioma–parenchymal crosstalk. Autocrine signalling plays a role in meningioma behaviour, but it is unknown how paracrine signalling between tumour cells and glia cells takes place. Interleukins are important regulators of brain inflammation [13], and it is interesting to note that the pattern of cytokine expression in meningiomas largely resembles gliomas [38]. Cytokines in the glioma microenvironment mediate interactions between tumour cells and glial cells [13], and it is likely that these cytokines mediate similar crosstalk in meningiomas. The relationship between inflammatory cell infiltration, the expression of cytokines and its association with meningioma brain invasion has not been investigated yet. It is noteworthy that other cells in the TME also might interact with the brain parenchyma. For instance, mast cells, which are known to infiltrate meningiomas, have been suggested to participate in neuroinflammation directly and through microglia stimulation in a range of CNS pathologies [85]. These interactions, however, are out of scope for this review. The TME in meningiomas may have several possible targets for immunotherapy and tolerogenic therapy, but further basic research in this field is needed. Programmed Cell Death Protein 1 (PD-1/PD-1L) expression is the best documented immunosuppressive mechanism in meningiomas [77,86,87] but has only been evaluated in the treatment of other brain tumours rather than meningiomas [88]. Taking the M1/M2 ratio into account may be beneficial in the grading and prognostication of meningiomas, and rendering the ratio towards a tumoricidal M1 expression could perhaps resemble a therapeutic target for WHO grade II and III meningiomas. Targeting cytokines involved in the meningioma–brain crosstalk is also a subject of debate; however, the effect of a single cytokine can be pleiotropic depending on the concentration and exposure time, and most single cytokines, notably of the IL-1 family, can have either pro- or anti-tumoral effects [89], which is why a better understanding of the crosstalk is needed in order to suggest any targets. Furthermore, the role of radiation in modulating the inflammatory response and the TME in meningiomas would be of great interest and is highly topical for targeted and immuno-therapeutic treatments for other types of cancer [90].

### 4.2. Research Gaps and Future Research Recommendations

In this work, we defined two preconditions necessary for reciprocal tumour–brain crosstalk: (1) attraction and activation of glial cells from the parenchyma to the tumour site and (2) the reciprocal exchange of substances between glial cells and tumour cells that impact meningioma growth and behaviour.

Next, we used this framework to investigate if the literature described such elements of crosstalk in meningioma biology. Based on our search, it is still uncertain to what extent these preconditions are present. Microglia play a central role in tumour–brain crosstalk in other primary brain tumours [13,21]. To explore if meningiomas elicit an inflammatory response in the brain parenchyma, it is essential to establish whether the mononuclear cells in the TME constitute native microglia, blood-derived macrophages or a combination of the two. Functional differences exist between the two cell populations, at least in glioblastoma multiforme [84], underlining the relevance of discriminating between the two cell types. This is a challenge due to the heterogeneity of microglia and the similarities between microglia and macrophages. The P2RY12 receptor and TMEM119 have recently been outlined as specific microglia markers [27]. The activation of glial cells probably takes place at the tumour–brain interface of invasive meningiomas, and, therefore, future studies should focus on the brain parenchyma adjacent to invasive meningiomas. First, the tumour–brain interface needs further characterisation by immunohistochemical techniques. Second, the possible reciprocal interactions between tumour cells and glia cells should be explored.

One of the main barriers to understanding meningioma biology is the lack of in vitro models that faithfully mimic in vivo conditions [91]. Limitations in existing in vitro models include cell culture conditions that differ significantly from the TME and changes in tumour cell physiology due to immortalisation [91]. Co-cultures of brain organoids with cells from glioblastomas have recently successfully modelled glioblastoma growth in vitro, providing a path towards more accurate in vitro models of meningiomas [91,92]. Genetically based and xenograft mouse models, which recapitulate meningiomas in vivo, overcome many of the limitations of in vitro models. Such models have been used to investigate microglia/astrocyte–glioblastoma interactions [91,93,94]. Nonetheless, available in vivo models of meningiomas are associated with several shortcomings. Genetically engineered mouse models of meningiomas are less scalable, more costly, lack tumour heterogeneity and are associated with non-specific effects of the genetic alterations. Available xenograft models have the advantage of using actual human tumour cells but are costly and technically complicated. In addition, the use of immunocompromised mice implies abnormal tumour immunology and microenvironment [31,91]. Organotypic brain slice cultures enable the evaluation of interactions between tumour cells and the brain parenchyma with a preserved cytoarchitecture of the brain. The main limitation of this method is the short duration of viable cultures, which is problematic when modelling slow growing tumours, such as meningiomas [31]. In summary, expanding the knowledge on interactions between tumour cells and the brain parenchyma is dependent on the improvement of preclinical models that successfully mimic the complexity of the TME.

## 5. Limitations of This Review

Meningioma–brain crosstalk has not been directly assessed before, which, of course, imposes possible limitations. Choosing search terms that successfully cover our topic poses a challenge. We chose search terms to cover tumour cells, glia cells and cytokines, as they are key mediators of crosstalk in other brain tumours [13]. There are, however, other potential mediators of tumour–brain crosstalk that were not incorporated in our search strategy, leading to the risk of selection bias. Moreover, applying microglia/astrocyte–glioblastoma interactions as a proxy of tumour–brain crosstalk should be carried out with caution, as meningiomas and glioblastomas are tumours of different origin and biology. A major limitation is the low number of available studies on meningioma–brain interactions. Hence, a systematic review was not feasible, and we synthesised the available information with a scoping review. Given that the scope of this review is largely unexplored, some sections are based on very few studies, and the included studies have not undergone a quality assessment. This in combination constitutes a potential risk of overinterpreting the results. Other limitations include that we only searched the PubMed database, did not include unpublished results and only included research published in the English language.

## 6. Conclusions

The role of the brain parenchyma in the pathogenesis of meningiomas is largely unexplored. In this scoping review, we summarised and disseminated research findings that might elucidate whether tumour crosstalk with the brain parenchyma plays a role in meningioma growth and behaviour. It is still questionable to what extent meningiomas elicit an immune response in the brain. Furthermore, the extent and type of immune response seem to vary amongst tumours. Phagocytes are frequently detected as the most abundant infiltrating immune cells in meningiomas, but it seems reasonable that these are blood-derived macrophages rather than resident microglia. As the pial–glial basement membrane might protect the brain parenchyma adjacent to meningiomas from inflammation, it is possible that tumour–brain crosstalk is of the highest significance in meningiomas with disruption of this membrane. We suggest that future research should focus on the inflammatory response at the tumour–brain interface of invasive meningiomas, as these cells are more likely to represent activated glia cells from the brain parenchyma.

## Figures and Tables

**Figure 2 cancers-13-04267-f002:**
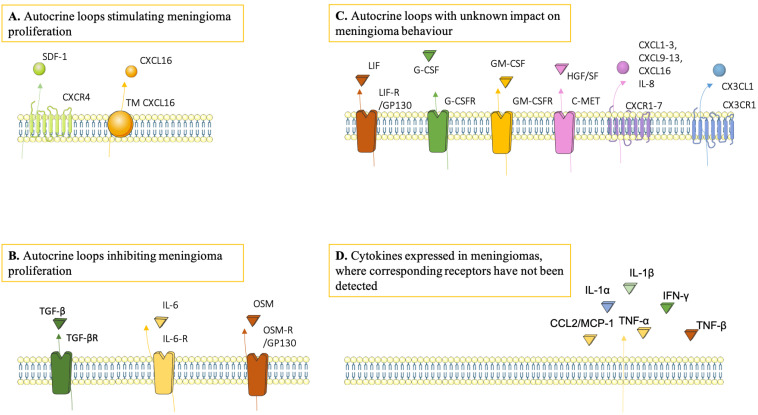
Simultaneous expression of cytokine/cytokine receptor pairs in meningiomas, suggestive of autocrine loops. Some of these autocrine loops showed impact on meningioma growth in vitro. (**A**) SDF-1/CXCR4 and CXCL16/TM-CXCL16 signalling stimulated meningioma proliferation and growth. (**B**) TGF-ß/TGF-ßR, IL-6/IL-6R and OSM/OSM-R signalling inhibited meningioma proliferation and growth (**C**) Loops of which the consequence on meningioma proliferation/growth have not been investigated in vivo. (**D**) Cytokines expressed in meningiomas, where corresponding receptors have not been detected.

**Table 1 cancers-13-04267-t001:** Overview of cytokines and cytokine receptors expressed in meningiomas ranging from WHO grade I–III. Cytokines are key mediators of crosstalk in other primary brain tumours. A better understanding of the cytokine microenvironment is essential to explore the role of the brain parenchyma in the pathogenesis of meningiomas. “+” indicates that the receptor has been detected in meningioma tumour cells. Abbreviations: n.a. = not applicable.

Cytokine	n. Positive Meningiomas/Total	Cytokine Receptors	+/n.a.
Type 1 Cytokine Family			
IL-2	0/5 [38], 0/9 [39], 0/11 [40]	IL-2R	n.a.
IL-3	11/11 [40]	IL-3R	n.a.
IL-4	0/11 [40], 0/5 [38], 0/9 [39]	IL-4R	+ [41,42]
IL-5	0/11 [40], 1/5 [38]	IL-5R	n.a.
IL-6	11/11 [40], 1/5 [38], 10/10 [43], 9/9 [39], 0/2 [44], 10/10 [45]	IL-6R	+ [43]
IL-7	0/11 [40]	IL-7R	n.a.
G-CSF	28/30 [46]	G-CSFR	+ [46]
GM-CSF	1/5 [38], 27/30 [46]	GM-CSFR	+ [46]
Type 2 Cytokine Family			
IFN-γ	11/11 [40]	IFN-γR	n.a.
IL-10	1/5 [38]	IL-10R	n.a.
OSM	2/2 [44], 10/10 [43]	OSM-R	+ [43]
LIF	2/2 [44], 10/10 [43]	LIF-R	+ [43]
TNF Cytokine family			
TNF-α	3/5 [38], 0/11 [40]	TNFR1	n.a.
TNF-*β*	6/11 [40]	TNFR1	n.a.
IL-1 Family			
IL-1α	0/11 [40], 5/5 [38]	IL-1R	n.a.
IL-1*β*	5/11 [40], 3/5 [38], 9/9 [39]	IL-1R	n.a.
Other Cytokines			
TGF-α	22/26 [47]	EGFR	+ [47,48,49]
TGF-*β1*	8/11 [40], 3/5 [38], 3/6 [50]	TGF-*βRI-III*	+ [51]
TGF-*β2*	9/11 [40], 5/5 [38], 6/6 [50]	-	-
TGF-*β3*	11/11 [40], 5/5 [38], 6/6 [50]	-	-
HGF/SF	5/14 [52]	MET	+ [52,53]
Chemokines			
IL-8	11/11 [40], 14/35 [54]	CXCR1/2	+ [54]
CCL2/MCP-1	16/16 [18]	CCR2	n.a.
CXCL1/2/3	27/27 [54]	CXCR1/2	+ [54]
CXCL9	10/36 [54]	CXCR3	+ [54]
CXCL10	15/36 [54]	CXCR3	+ [54]
CXCL11	22/22 [55]	CXCR3	+ [54]
CXCL12/SDF-1	22/22 [55], 6/6 [56], 29/55 [54]	CXCR4/CXCR7	+ [25,55,56,57]
CXCL13	1/35 [54]	CXCR5	+ [54]
CXCL16	27/27 [58], 28/28 [59]	CXCR6/tmCXCL16	(+)/+ [58,59]
CX3CL1	27/27 [58]	CX3CR1	+ [58]

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
