# Peer review of "Meningioma–Brain Crosstalk: A Scoping Review"

_cancers, 2021, doi:10.3390/cancers13174267_

Round 1

Reviewer 1 Report

The review proposed by authors may be of interest to people working in the meningioma field, mostly. When referring to Appendix 1 in line 107  I guess you mean the SM file, which has not been properly transformed into the pdf file, since some text seems to be cut or unreadable. Please fix that.

Author Response

Thank you very much for your thorough feedback, questions and suggestions.

We have tried to implement all to the best of our capabilities. We have left all changes transparent in word track-changes.

Reviewer #1:

The review proposed by authors may be of interest to people working in the meningioma field, mostly. When referring to Appendix 1 in line 107  I guess you mean the SM file, which has not been properly transformed into the pdf file, since some text seems to be cut or unreadable. Please fix that.

  • The table has now been fixed.

Reviewer 2 Report

Title: “Meningioma-Brain Crosstalk: A Scoping Review”

The authors present a highly interesting review focusing on the role of tumor microenvironment (TME) in meningioma. As important as this aspect is, as little evidence has been created shedding light into the crosstalk between meningioma and the surrounding host organ. Therefore, this review is very relevant to the audience, and there are only a few aspects, which need to be addressed before being published in Cancers.

  1. Simple summary: Strictly speaking, meningiomas are not brain tumors. Since they arise from the meningothelial cap cells and therefore being from both mesodermal and neural origin, they are intracranial tumors but not primary brain tumors. In addition, they are in fact the most frequent primary intracranial tumor. It would be great if this sentence could be changed accordingly (as correctly stated in the introduction…).
  2. Simple summary: What exactly is indicated by the sentence “the treatment remains simplistic…”. From a neurosurgical standpoint, meningioma treatment is far from simplistic….
  3. Simple summary: How exactly is the treatment “…to some extent insufficient compared to other brain tumors” ? The wording should be clearer and more precise.
  4. Discussion: The authors state that tumor associated astrocytes “disappeared” during the process of brain invasion. What exactly does this indicate ? A reduction of astrogliosis in brain invasive meningiomas ? Is this a continuous process during invasion ? This would be highly interesting as the meningioma cells would have to penetrate a microglial and astroglial border / wall to invade the brain. Pls. specify this important aspect.
  5. The aspect of paracrine signaling from the TME to the meningioma appears highly important. Is there any evidence that an increased inflammatory cell infiltrate or a higher expression of inflammatory cytokines is associated with increased meningioma cell invasion into the brain ?
  6. Is there any evidence linking extent of resection as well as oncological outcome with TME dynamics in meningioma ? For example, is a higher degree of inflammatory infiltrate correlated to worse outcome ? I am not aware of such evidence, however it would additionally strengthen the review if the impact of TME on oncological outcome would be discussed.

Author Response

Thank you very much for Your thorough feedback, questions and suggestions.

We have tried to implement all to the best of our capabilities. We have left all changes transparent in word track-changes.

Reviewer #2:
Title: “Meningioma-Brain Crosstalk: A Scoping Review”

The authors present a highly interesting review focusing on the role of tumor microenvironment (TME) in meningioma. As important as this aspect is, as little evidence has been created shedding light into the crosstalk between meningioma and the surrounding host organ. Therefore, this review is very relevant to the audience, and there are only a few aspects, which need to be addressed before being published in Cancers.

  1. Simple summary: Strictly speaking, meningiomas are not brain tumors. Since they arise from the meningothelial cap cells and therefore being from both mesodermal and neural origin, they are intracranialtumors but not primary brain tumors. In addition, they are in fact the most frequent primary intracranial tumor. It would be great if this sentence could be changed accordingly (as correctly stated in the introduction…).
    => Thanks. We agree. The sentence has been changed according to the suggestion.

  1. Simple summary: What exactly is indicated by the sentence “the treatment remains simplistic…”. From a neurosurgical standpoint, meningioma treatment is far from simplistic….
    => Thanks for pointing this out. The sentence has been changed, what we actually meant was the diagnostics…

  1. Simple summary: How exactly is the treatment “…to some extent insufficient compared to other brain tumors” ? The wording should be clearer and more precise.
    => The sentence has been changed, what we actually meant was the diagnostics.

  1. Discussion: The authors state that tumor associated astrocytes “disappeared” during the process og skriv her explicit hvilken text some r ændred / addedof brain invasion. What exactly does this indicate ? A reduction of astrogliosis in brain invasive meningiomas ? Is this a continuous process during invasion ? This would be highly interesting as the meningioma cells would have to penetrate a microglial and astroglial border / wall to invade the brain. specify this important aspect.

=> The authors of the article (Zeltner et al 2007) interpreted that astrocytes “disappear”, but they only described that astrocytes were absent from the pial border between meningioma cells and parenchyma in cases where the basal membrane were absent. We have rephrased this to:
“In specimens with brain invasion, focal absence of a pial basal membrane correlated with absence of subpial astrocytes[34]. The authors interpreted the finding as “astrocytic disappearance” after invasion and suggested that astrocyte survival was dependent on adherence to an intact basal membrane Loss of astrocytes would then be a result of a degraded basal membrane. In analogy, breaches of the basal membrane are considered hallmarks of malignancy in cancers[74]. Degradation of the basal membrane could be mediated by matric metalloproteases, that are expressed by tumor cells in invasive and aggressive meningiomas[75,76]. Both the known co-expression of MMP-9 with MIF-1[77] and immune regulatory T-cells would further allow brain invasion by attenuation of the microglial response[78]. Although brain invasion in meningioma not necessarily leads to the diagnosis of WHO grade III meningioma, brain infiltration still appears to be associated with aggressive tumor behavior [2,10,79–81]

  1. The aspect of paracrine signaling from the TME to the meningioma appears highly important. Is there any evidence that an increased inflammatory cell infiltrate or a higher expression of inflammatory cytokines is associated with increased meningioma cell invasion into the brain ?
    => This is a very relevant and important question. Unfortunately, the studies we have found and included in the paper does not investigate this. We have addressed this with the following sentence:
    “The relationship between inflammatory cell infiltration, the expression of cytokines and its association to meningioma brain invasion has not been investigated yet.”

  1. Is there any evidence linking extent of resection as well as oncological outcome with TME dynamics in meningioma ? For example, is a higher degree of inflammatory infiltrate correlated to worse outcome ? I am not aware of such evidence, however it would additionally strengthen the review if the impact of TME on oncological outcome would be discussed.
    => In regard to the TME dynamics, not that we are aware of either. JLHV is currently doing a PhD on this specific subject. We added to the results section – Classical (M1) or alternative (M2) microglia/macrophage activation in meningiomas:
    “Proctor et al.(Proctor et al. 2019) found that infiltration of TAM’s increased with WHO grade and the M1:M2 cell ratio was significantly decreased in WHO grade II, compared to grade I tumors, but the relationship between the post-operative outcome and the TME dynamics in regard to extent of inflammation remains to be explored.”

Reviewer 3 Report

The article entitled: “Meningioma-Brain Crosstalk: A Scoping Review” is written very comprehensively. It is easy to read and leads the reader through the presented topic step by step.

Below please find some advice, which in my opinion will improve the article:

- Table 1: the last column “y/n.a.” is unclear. Pleas make this information more understandable.

- PRISMA diagram – something went wrong during the conversion of the chart. Some information is cut.

Author Response

Thank you very much for Your thorough feedback, questions and suggestions.

We have tried to implement all to the best of our capabilities. We have left all changes transparent in word track-changes.

Reviewer #3:
The article entitled: “Meningioma-Brain Crosstalk: A Scoping Review” is written very comprehensively. It is easy to read and leads the reader through the presented topic step by step.

Below please find some advice, which in my opinion will improve the article:

- Table 1: the last column “y/n.a.” is unclear. Please make this information more understandable.
=> The table has been and further explained in the text below.

- PRISMA diagram – something went wrong during the conversion of the chart. Some information is cut.
=> The diagram has been fixed.

Reviewer 4 Report

This is a concise review on TME in meningioma focusing on two specific modes of cell-cell interaction. The manuscript is written well. Minor aspects can be addressed to enhance overall quality:

1.) l. 221 ff. - the authors state that infiltration is a hallmark of malignancy. Brain invasion in meningioma does not necessarily lead to the diagnosis of anaplastic meningioma. The statement needs further clarification.

2.) It is well accepted that meningioma is not a homogenous entity. If the authors encountered differences of TME interactions across meningioma locations (skull base versus convexity, parasagittal, etc.) this would be worth mentioning.

3.) The reader would appreciate the authors' perspective on potential therapeutic targets. One would think that the TME may represent a promising field for pharmaceutical interventions. 

Overall, the work is a great foundation to help orientate colleagues interested in researching the TME in meningiomas.  

Author Response

Thank you very much for your thorough feedback, questions and suggestions.

We have tried to implement all to the best of our capabilities. We have left all changes transparent in word track-changes.

Reviewer #4:
This is a concise review on TME in meningioma focusing on two specific modes of cell-cell interaction. The manuscript is written well. Minor aspects can be addressed to enhance overall quality:

1.) l. 221 ff. - the authors state that infiltration is a hallmark of malignancy. Brain invasion in meningioma does not necessarily lead to the diagnosis of anaplastic meningioma. The statement needs further clarification.
=> We have rephrased this to:
“In specimens with brain invasion, focal absence of a pial basal membrane correlated with absence of subpial astrocytes[34]. The authors interpreted the finding as “astrocytic disappearance” after invasion and suggested that astrocyte survival was dependent on adherence to an intact basal membrane Loss of astrocytes would then be a result of a degraded basal membrane. In analogy, breaches of the basal membrane are considered hallmarks of malignancy in cancers[74]. Degradation of the basal membrane could be mediated by matric metalloproteases, that are expressed by tumor cells in invasive and aggressive meningiomas[75,76]. Both the known co-expression of MMP-9 with MIF-1[77] and immune regulatory T-cells would further allow brain invasion by attenuation of the microglial response[78]. Although brain invasion in meningioma not necessarily leads to the diagnosis of WHO grade III meningioma, brain infiltration still appears to be associated with aggressive tumor behavior [2,10,79–81]

2.) It is well accepted that meningioma is not a homogenous entity. If the authors encountered differences of TME interactions across meningioma locations (skull base versus convexity, parasagittal, etc.) this would be worth mentioning.
=> This is a very relevant inquiry. Unfortunately, granular data is lacking. We made no changes to the manuscript.

3.) The reader would appreciate the authors' perspective on potential therapeutic targets. One would think that the TME may represent a promising field for pharmaceutical interventions.> We have added a brief speculation on potential targets. Please see following text:
“The TME in meningiomas may have several possible targets for immunotherapy and tolerogenic therapy, but further basic research in this field is needed. Programmed Cell Death Protein 1 (PD-1/PD-1L) expression is the best documented immunosuppressive mechanism in meningiomas [78,87,88] but has only been evaluated in the treatment of other brain tumors than meningiomas[89]. Taking the M1/M2 ratio in to account may be beneficial in grading and prognostication of meningiomas and rendering the ratio towards a tumoricidal M1-expression could perhaps resemble a therapeutic target for WHO grade II and III meningiomas. Targeting cytokines involved in the meningioma-brain crosstalk is also a subject of debate, however the effect of a single cytokine can be pleiotropic depending on the concentration and exposure time and most single cytokines, notably of the IL-1 family, can have either pro- or anti-tumoral effects [90] why better understanding of the crosstalk is needed in order to suggest any targets.
Furthermore the role of radiation in modulating the inflammatory response and the TME in meningiomas would be of great interest and is highly topical for targeted and immuno-therapeutic treatments for other types of cancer
[91].”

Overall, the work is a great foundation to help orientate colleagues interested in researching the TME in meningiomas.